# Endogenous Extracellular Matrix Regulates the Response of Osteosarcoma 3D Spheroids to Doxorubicin

**DOI:** 10.3390/cancers15041221

**Published:** 2023-02-14

**Authors:** Margherita Cortini, Francesca Macchi, Francesca Reggiani, Emanuele Vitale, Maria Veronica Lipreri, Francesca Perut, Alessia Ciarrocchi, Nicola Baldini, Sofia Avnet

**Affiliations:** 1Biomedical Science and Technology and Nanobiotechnology Laboratory, Istituto di Ricovero e Cura a Carattere Scientifico, IRCCS Istituto Ortopedico Rizzoli, 40136 Bologna, Italy; 2Department of Biomedical and Neuromotor Sciences, Alma Mater Studiorum, Università di Bologna, 40127 Bologna, Italy; 3Laboratory of Translational Research, Azienda Unità Sanitaria Locale—Istituto di Ricovero e Cura a Carattere Scientifico, IRCCS di Reggio Emilia, 42123 Reggio Emilia, Italy; 4Clinical and Experimental Medicine PhD Program, University of Modena and Reggio Emilia, 41125 Modena, Italy

**Keywords:** osteosarcoma, 3D models, extracellular matrix, collagen, spheroids, tumor microenvironment, drug screening

## Abstract

**Simple Summary:**

The pathogenesis of osteosarcoma relies on complex interactions between developing cancer and surrounding tissue, which includes proteins of the extracellular matrix. Mapping ECM–cell interactions and ECM composition is highly important to understand and predict cancer response to chemotherapy and potentially give rise to alternative targets for therapy. Our study aims at generating a 3D model that recapitulates interactions of cancer cells with ECM components and with non-tumor stromal cells and at elucidating the role of ECM deposition in chemotherapy response. Dissecting the contribution of the tumor environment and the role of collagenic and non-collagenic proteins of the ECM will provide additional knowledge for the development of new antitumor strategies.

**Abstract:**

The extracellular matrix (ECM) modulates cell behavior, shape, and viability as well as mechanical properties. In recent years, ECM disregulation and aberrant remodeling has gained considerable attention in cancer targeting and prevention since it may stimulate tumorigenesis and metastasis. Here, we developed an in vitro model that aims at mimicking the in vivo tumor microenvironment by recapitulating the interactions between osteosarcoma (OS) cells and ECM with respect to cancer progression. We long-term cultured 3D OS spheroids made of metastatic or non-metastatic OS cells mixed with mesenchymal stromal cells (MSCs); confirmed the deposition of ECM proteins such as Type I collagen, Type III collagen, and fibronectin by the stromal component at the interface between tumor cells and MSCs; and found that ECM secretion is inhibited by a neutralizing anti-IL-6 antibody, suggesting a new role of this cytokine in OS ECM deposition. Most importantly, we showed that the cytotoxic effect of doxorubicin is reduced by the presence of Type I collagen. We thus conclude that ECM protein deposition is crucial for modelling and studying drug response. Our results also suggest that targeting ECM proteins might improve the outcome of a subset of chemoresistant tumors.

## 1. Introduction

The tumor microenvironment (TME) is a key factor for cancer development and malignancy. It is composed of different cell types, and the interplay among these cells determines the production of several soluble factors and components of the extracellular matrix (ECM) with the ability to modulate the growth and aggressiveness of solid tumors [1,2]. In this view, the reactive stroma is now considered a foe in the development and progression of cancer [3,4,5]. In addition to the cellular components, the TME is also composed of a three-dimensional network of ECM proteins, which not only serve as a scaffold for tumor cell growth but also regulate cell–cell or cell–matrix crosstalk, in turn affecting the ability of tumor cells to spread and metastasize. The ECM also influences cell differentiation, proliferation, and homeostasis [6] and modulates the response to anticancer drugs [7,8]. Each tumor isotype has a peculiar ECM composition and whether the presence of collagen-rich ECM is pro- or antitumorigenic is still controversial. Indeed, the ECM can act as a physical barrier to decrease tumor perfusion and drug delivery and facilitates cell migration and tumor aggressiveness, especially in the presence of heavily cross-linked and linearized collagens [7,9,10,11]. The role of the ECM is also isotype- and location-dependent [10,12], and, within the same tumor, the expression of ECM proteins may differ in the primary versus the metastatic site. Cancer cells escaping the primary tumor mass might be facilitated by the presence of the ECM, which favors infiltration and migration by offering integrins and adhesion structures as a track for migrating cells, or can oppositely be surrounded by the ECM barrier [9,10]. These two apparently opposite phenomena are strictly isotype- and context-dependent. Regarding the metastatic niche, distant sites targeted by metastatic cells may be influenced both by the remodeling of ECM through the deposition of proteins, e.g., fibronectin and collagen, that may facilitate the engraftment of tumor cells [13], and by circulating soluble factors that may be released by the primary tumor. All these changes prime the initially healthy organ microenvironment and render it amenable for subsequent metastatic cell colonization [14]. Thus, in light of the existing data, ECM should be always included in in vitro cancer models and for drug screening. In recent decades, significant improvements have been made in cell culture methods, and the scientific community is nowadays aligned with the idea that an ideal cell culture model should not only mimic oncogenesis and cell proliferation, but also imitate the interactions between cells intermingled with the ECM [6,15]. Furthermore, in addition to cancer cells, to enhance ECM production and to better mimic TME, these advanced cancer models should include mesenchymal stromal cells (MSCs). Indeed, MSCs are physiologically imprinted cells that home to areas of insults and inflammation, including cancer, and bear the ability to modulate the TME and ECM deposition. Upon activation in the presence of tumor cells, stromal cells may also change their secretory phenotype and they have been already demonstrated to play a major role in the context of sarcoma genesis [2,16]. However, the role of ECM secretion by MSCs in the OS TME has never been explored yet. Due to these features, MSCs might therefore play a direct role in the development of the tumor primary site and the metastatic niche [17], as well as in regulating drug response.

We focused on osteosarcoma (OS), a very aggressive malignancy of mesenchymal origin that arises in bone. In OS, the TME has been associated with tumor aggressiveness and resistance to antineoplastic drugs [3,18], and, in particular, in preclinical models, the ECM has been demonstrated to enable a supportive scaffold for OS progression [19,20,21]. Type I collagen has been shown to increase the synthesis and activation of MMP-2, which in turn promotes OS progression and metastasis [22,23]. Type III collagen has instead been associated with resistance to methotrexate [11], suggesting a poor diffusion of the drug in the presence of a fibrillary insoluble matrix. On the other hand, fibronectins display functional motifs that interact with integrins [19]; selective down-regulation of integrins in OS has resulted in the decreased deposition of fibronectin and has been associated with a reduced cell adhesion and increased spread [24]. Conversely, in other studies, upregulation of integrins has been shown to enhance the adhesiveness of OS cells to fibronectin [25] and an increase in fibronectin expression has been associated with chemoresistance [26].

Bearing in mind that 2D monolayer cell cultures may not be representative of in vivo conditions, in this study, we generated 3D spheroids of metastatic and non-metastatic OS cell lines mixed with MSCs, with the aim of characterizing the endogenous secretion and deposition of ECM proteins and laying the basis for pharmacological drug screening. In our 3D OS models, we demonstrated that the presence of MSCs increases ECM protein deposition via IL-6 and that Type I collagen expression is directly linked to the aggressiveness of OS cells by modulating the chemotherapeutic response to doxorubicin (DXR).

## 2. Materials and Methods

### 2.1. Cell Culture

ADMSC-GFP cells were kindly provided by Professor Massimo Dominici (Modena, Italy); the transfection protocol was previously described [27]. 143B (ATCC, #CRL-8303), MG-63 (ATCC, #CRL-1427), and Saos-2 (ATCC. #HTB-85) cell lines were purchased from the American Type Culture Collection (ATCC, Washington, DC, USA) and cultured in IMDM (Life Technologies, Carlsbad, CA, USA) plus penicillin (20 U/mL), streptomycin (100 mg/mL), and 10% heat-inactivated fetal bovine serum for a range of 10–20 passages from thawing. Cells were maintained at 37 °C in a humidified atmosphere with 5% CO_2_. All cell lines were tested against mycoplasma with nuclear staining every month.

For 3D models, to form spheroids, RPMI was added with sodium bicarbonate and adjusted to pH 7.4. Spheroids were thus grown for 3, 7, or 14 days, according to experimental needs. We obtained hanging-drop spheroids as previously described [18]. For homotypic spheroids, 5 × 10^3^ OS cells were plated in a 96-well round-bottom ultra-low-attachment plate (Costar, WA, USA), whereas heterotypic tumor/stroma spheroids were seeded in a 1:3 ratio (5 × 10^3^ OS cells mixed with 1.5 × 10^4^ ADMSC-GFP). The plate was flipped and incubated in gentle shaking at 37 °C and 5% CO_2_ overnight. The following day, the plate was flipped again, 200 μL of medium was removed, and the time point was indicated as T0; the spheroids were then let grow for additional 3, 7, or 14 days.

For the calculation of DXR IC50 in 3D, hanging-drop spheroids were formed as described. DXR was added at increasing doses and Alamar Blue viability was assessed after 96 hrs. For IC50 in 2D, cells were cultured in monolayers, grown for 96 h, and used for Alamar staining (ThermoFisher, Waltham, MA, USA).

### 2.2. Immunofluorescence

For immunofluorescence staining, spheroids were grown for 3, 7, or 14 days, then were coated with a 2% agarose layer and included in OCT (TissueTek, Alphen aan den Rijn, The Netherlands). Spheroids were fixed with 3.7% paraformaldehyde for 20 min and next incubated with anti-human collagen Type III (FH-7A, Abcam, #ab6310, Cambridge, UK), anti-human collagen Type I (C-11, #Mab 1340, Merck, Darmstadt, Germany), and anti-human fibronectin (IST-3, #MAB1892, Merck) monoclonal antibodies. Primary antibodies were followed by staining with secondary Alexa647 antibodies (1:250, #A11011, Life Technologies). Nuclei were counterstained with Hoechst 33258 (0.125 μg/mL, #H3569, ThermoFisher). Images were acquired with an air objective 20×, numerical aperture 0.75, Galvano scanning, zoom at 1, and line average of 4 (A1R MP confocal microscope, Nikon, Tokyo, Japan).

For live acquisition, spheroids were stained overnight with anti-human collagen Type I and then for 2 h with Hoechst 33342 (0.5 μg/mL, #b2261, Merck). Images were acquired with a 25× water-immersion objective and multi-photon excitation fluorescence, resonant scanning, zoom at 1, and line average of 4 for a total of 100 slices on the Z-stack (A1R MP confocal microscope, Nikon).

### 2.3. Immunohistochemistry

Canine OS paraffin-embedded samples were obtained from the Department of Veterinary Sciences, University of Bologna, Bologna, Italy (authorization from the Ministry of Health n. 0008868-03/04/2015-DGSAF-COD_UO-P) at the time of the surgical treatment (amputation) before adjuvant chemotherapy.

Paraffin-embedded mouse xenografts were previously obtained [27]. For both subcutaneous and orthotopic models, we used 5-week-old male NOD/SCID mice (Charles River Laboratories International, Wilmington, MA, USA). For subcutaneous models, we randomly split the animals into two groups for the subcutaneous injection in the flank of homotypic or heterotypic cell populations, mixed with reduced growth factor Matrigel^®^ (BD Life Sciences, Biosciences, Franklin Lakes, NJ, USA). Weights were taken daily during treatment. For orthotopic models we randomly split the animals into two groups and injected them with luc-143B cells, without ADMSC-GFP, in the left tibia. Cells were suspended in an isotonic saline solution. As previously described [27], 10 μL of cell suspension was slowly injected into the medullary cavity of the left tibia. The micro-syringe was then removed, and bone wax was used to seal the hole. In all cases, mice were euthanized when the tumor volume exceeded 2500 mm^3^. All the procedures involving the animals were conducted according to national and international laws on experimental animals (L.D. 26/2014; Directive 2010/63/EU) and the approved experimental protocol procedure (approved by the Ministry of Health, protocol n. 393/2015-PR of 20 May 2015).

Representative 5 µm thick sections were mounted on a glass slide covered with 2% silane solution in acetone. After dewaxing in HistoClear (HistoLine Laboratories, Milan, Italy) and rehydration in ethanol, staining was performed with EnVision Flex, High pH (Agilent, Dako Omnis) according to the manufacturer’s instructions. Briefly, the slides were incubated for 30 min at 98 °C in a high-pH antigen-retrieval solution. After cooling, the sections were incubated in peroxidase blocking solution and non-specific binding was blocked with incubation in 5% bovine serum albumin. The following primary antibodies were used: anti-human Ki67 (clone MIB-1, #M7240, Agilent, Santa Clara, CA, USA) and anti-human collagen Type I. Sections were developed with DAB and counterstained with Mayer’s hematoxylin. Negative controls were also performed by omitting the primary antibody.

To separate GFP-positive ADMSCs from heterotypic MG63/MSC spheroids, cell aggregates were harvested after 14 days of culture growth, digested with a Tumor Dissociation Kit (Miltenyi Biotec, Bergisch Gladbach, Germany) for 30 min at 37 °C, strained with a 100 μm strainer (StarLab, Milan, Italy), washed with IB buffer (1% BSA, 5 mM EDTA in PBS), and resuspended in IB buffer to a final concentration of 1 × 10^6^ cells/mL. GFP-positive and GFP-negative populations were separated using a FACS Melody sorter (BD, Franklin Lakes, NJ, USA) with forward and side-scatter gating and GFP expression (FITC channel). A small sorted population was used to check cell viability with forward and GFP gating.

### 2.4. RNA Isolation and Gene Expression

RNA was extracted from sorted cells with Trizol reagent (ThermoFisher) using glycogen as a carrier to increase the yield. The total RNA was reverse transcribed into cDNA using RNase inhibitor and MuLV Reverse Transcriptase (Applied Biosystems, Foster City, CA, USA). First-strand cDNA was synthesized with RT-qPCR using random hexamers. Real-time polymerization chain reaction (real-time PCR) was performed by amplifying 500 ng using the SsoAdvanced Sybr Green Mix (Biorad, Hercules, CA, USA) and the CFX96Touch instrument (Biorad). Relative gene expression was obtained with the ratio to GAPDH expression with the following sequences: GAPDH For: ccaaggagtaagacccctgg; GAPDH Rev: aggggagattcagtgtggtg; collagen Type I For gctcactttccaccctctct; collagen Type I Rev: ttcagaggagagaggtcgga; collagen Type III alpha I For: aagaaggccctgaagctgat; collagen Type III alpha I Rev: gtgtttcgtgcaaccatcct; fibronectin For: tccccaactggtaacccttc; fibronectin Rev: tgccaggaagctgaatacca.

### 2.5. Anti-IL-6 Assay

To evaluate collagen Type I deposition dependency from IL-6, neutralizing anti-IL-6 monoclonal antibody (Tocilizumab, Roche, Basel, Switzerland) was added to the spheroid medium at a final concentration of 100 μg/mL. Spheroids were repeatedly exposed to the antibody every 24 h for a total of 14 days. Spheroids were then live-stained for collagen Type I as described and acquired using multi-photon microscopy (Nikon). For each assay three replicates were performed. Quantification of collagen Type I was performed on the whole Z-stack (100 stacks/image) with NIS software (Nikon).

### 2.6. Collagenase and Doxorubicin Treatment

MG63/ADMSC-GFP spheroids were grown for 14 days as described to allow ECM deposition. After 14 days, spheroids were treated with collagenase IA (Merck) at 0.1 mg/mL or 1 mg/mL for 2 or 18 h. Spheroids were then washed, included in agarose and OCT, and sliced at 5 μm. Slices were stained for Type I collagen and counterstained with Hoechst 33258. At least 5 images/sample were acquired and Type I collagen was quantified with NIS software and normalized on the total number of nuclei.

To assess the effect of DXR, spheroids were grown for 14 days and then treated with collagenase IA 0.1 mg/mL for 18 h. At the end of collagenase treatment, DXR was added at 1 μM or 4.5 μM and spheroids were left in culture for an additional 72 h. Viability was then assessed with the Alamar Blue assay.

### 2.7. Statistical Analysis

Statistical analysis was performed with the GraphPad Prism 7 software (SAS Institute Inc.). The Mann–Whitney U test was used for unpaired comparison of two independent variables. Data were expressed as mean ± standard error (SE). Only *p* values < 0.05 were considered for statistical significance.

## 3. Results

### 3.1. Endogenous Deposition of ECM in OS Mixed Spheroids

To examine and characterize the presence of ECM proteins in OS spheroids, metastatic (143B) and non-metastatic (MG63 and Saos-2), and to assess whether the presence of GFP-transfected adipose-derived MSCs (ADMSC-GFP) results in increased ECM deposition, we first analyzed the presence of two abundantly expressed collagen isoforms in OS tissues, i.e., Type I collagen, Type III collagen, and fibronectin, using immunofluorescence (IF) after 3, 7, and 14 days of culture. As shown in Figure 1A and Appendix A and quantified in Figure 1B and Appendix A, ECM deposition increased consistently over time. Fibronectin precedes Type I and Type III collagen deposition. 143B spheroids expressed the lowest amount of ECM deposition, but showed a slight increase in Type III collagen and fibronectin deposition in the presence of ADMSC-GFP. MG63 and Saos-2 spheroids, instead, resulted in high expression levels of all three ECM proteins after 14 days of cell culture and in the presence of ADMSC-GFP.

IF staining of OCT-included spheroids showed the loss of the GFP signal from ADMSC-GFP cells. Therefore, to assess the spatial distribution of Type I collagen with respect to the localization of ADMSC-GFP cells, we performed live staining with multiphoton microscopy. In this analysis, we observed that ADMSC-GFP cells increased the IF signal of Type I collagen in all spheroids (Figure 2A, in red), in close proximity to the GFP signal of ADMSC-GFP cells (Figure 2A, images show bottom projection of the 3D rendering of the spheroids; Figure 2B, representative Z images of the live spheroids described in 2A).

The pseudo-color image of the signal corresponding to Type I collagen in representative sections of the spheroids also highlights an increased expression of this ECM protein in all tumor spheroids in the presence of ADMSC-GFP (Figure 2B).

### 3.2. ADMSCs Account for ECM Deposition in OS

To assess whether, in clinical samples, ECM protein deposition was mainly localized in the proximity of the stromal cells, we analyzed large tissue specimens of untreated spontaneous canine OS. Besides the existing technical limitation in distinguishing MSCs from OS cells, due to the shared origin and antigen profile expression of OS and MSCs, we focused on areas where bundles of reactive stroma were clearly identified (Figure 3A, compare small round tumor cells with fibroblast-like surrounding stromal cells). MSCs were positive for Type I collagen immunostaining (lower panel), trichromic staining (upper panel, in blue), and polarized light. Type I collagen was intensely positively birefringent with respect to the length of the fibers and appears with blue-purple staining; the positive intrinsic birefringence indicated an alignment parallel to the fiber. Conversely, Ki-67-positive cycling cells were prevalent in the tumor (see arrows) and did not show evidence of Type I collagen, either with immunohistochemistry or with polarized light (Figure 3A).

We also checked Type I collagen staining in mice subcutaneous (Figure 3B) and orthotopic xenografts models (Appendix A). For subcutaneous tumor growth, we co-injected ADMSC-GFP and 143B cells (with a 3:1 ratio) and sacrificed mice 22 days post-injection. Hematoxylin/eosin staining showed the presence of cells with a fibroblast-like morphology (Figure 3B, trichromic staining, black arrows highlight the structures that we identify as ADMSC-GFP) [27]. Polarized light and IHC showed the presence of Type I collagen in correspondence to fibroblast-like cells (see arrows), and IHC confirmed Ki-67 staining in tumor cells but not in elongated fibroblast-like cells. Trichromic staining could also be seen in 143B intratibial tumors (Appendix A), whereas a more intense staining was evident in organized fibrils in 143B+ADMSC orthotopic tumors (Appendix A).

As a proof-of-principle that ECM deposition in OS derives from ADMSC cells rather than tumor cells, we isolated the two cell populations from heterogeneous spheroids formed by MG63 and ADMSC-GFP cells. After 14 days of culture, spheroids were enzymatically digested and the single-cell suspension was analyzed with FACS and separated into GFP-positive (ADMSC-GFP) and GFP-negative (MG63) cell populations (Figure 4A). Despite the 3:1 ADMSC-GFP:MG63 ratio at the time of seeding, we could retrieve only 7.6% of ADMSC-GFP (98.4% pure population after cell sorting, Figure 4B), indicating that at the endpoint tumor cells take over the stromal cells (100% pure population after cell sorting, Figure 4B) with respect to T0 (Figure 4A, right panel). The two sorted populations were investigated for Type I collagen, Type III collagen, and fibronectin mRNA expression using real-time PCR and resulted in a prevalent ECM protein expression in MSCs, especially for Type III collagen (Figure 4C).

Overall, we could therefore demonstrate that ADMSC-GFP cells, physiologically imprinted for tissue regeneration, may also actively contribute to ECM deposition in OS tumors, and this phenotype can be reproduced in OS spheroids.

### 3.3. Type I Collagen Deposition in OS Spheroids Is Regulated by IL-6 Autocrine ADMSC Secretion

We have previously demonstrated that OS cells can reprogram MSCs to a tumor-associated phenotype [3], thus prompting MSCs to release a plethora of growth factors, cytokines, and chemokines that eventually support OS proliferation, migration, and stemness. Among these, tumor-stimulated MSCs secrete high levels of the pro-inflammatory cytokine IL-6 [3,28]. Here, we explored whether IL-6 might be responsible for the modulation of Type I collagen deposition. We performed live staining with multiphoton microscopy on MG63+ADMSC-GFP and 143B+ADMSC-GFP spheroids. Spheroids were exposed to the presence of a mAb blocking the IL-6 receptor/ligand interaction (i.e., Tocilizumab) and evaluated for Type I collagen expression with IF. Exposure of spheroids to the anti-IL-6 antibody significantly reduced the intensity of Type I collagen, as shown in Figure 5A and quantified in Figure 5B.

In summary, we showed that direct contact of OS with stromal cells increases the expression of ECM proteins, and in particular, that IL-6 contributes to the expression of Type I collagen. These features can be used to model the endogenous secretion of ECM in OS spheroids for mimicking in vivo TME and for drug screening.

### 3.4. ECM Deposition Influences Doxorubicin Cytotoxicity in OS Spheroids

To describe the role of endogenous ECM deposition in the modulation of the response to chemotherapy, we analyzed the effectiveness of DXR in OS cells cultured in 2D and 3D conditions in the presence or absence of MSCs. Increasing doses of DXR were tested in cells cultured as a 2D monolayer or as 3D spheroids and concentrations that lead to inhibition of 50% of the cells (IC50) are shown in Table 1. The IC50 values were significantly higher in spheroids rather than in monolayer cultures in MG63 and Saos-2 cell lines, whereas no differences were observed in low-secreting ECM 143B. The presence of ADMSC-GFP also contributed to increasing the IC50 in both MG63 and Saos-2 cells (Table 1), suggesting that the presence of ECM reduces the effectiveness of DXR against OS cells, possibly due to a lower penetration of the drug. Notably, the scarce ECM deposition observed in 143B spheroids did not significantly affect the IC50 values (Table 1).

To evaluate whether the observed IC50 differences reflected a different ECM deposition, we next treated OS spheroids (MG63+ADMSC-GFP) with collagenase IA (Figure 6A). We preliminarily assessed the duration and concentration of collagenase IA that leads to a significant decrease in Type I collagen deposition without affecting nuclear morphology (Figure 6A, representative images and Figure 6B quantification) and identified 0.1 mg/mL for 18 h as the optimal condition. At this dosage, collagenase IA did not affect collagen Type III (Appendix A).

Treatment of spheroids with a non-cytotoxic dose of collagenase IA prior to DXR treatment significantly improved DXR cytotoxicity, as assessed with an Alamar Blue viability assay (Figure 6C, * *p* < 0.05 at 4.5 μM vs. collagenase-untreated cells). Overall, these data indicate that Type I collagen deposition impairs anticancer drug effectiveness in 3D cultures, suggesting that that ECM inhibitors may be considered as candidate complementary therapeutic tools.

## 4. Discussion

3D cultures such as spheroids/tumoroids represent a promising new approach for preclinical drug screening [15]. In the long run, they will limit the use of animal models and improve prediction of the efficacy of anti-cancer drugs in patients. However, cancer is a highly heterogeneous disease and TME is complex and dynamic. The different features of TME must be included in the 3D model to recapitulate the pathophysiology of the tumor in vivo for an accurate evaluation of drug efficacy and toxicity. Spheroids are based on self-assembly involving cell aggregation and cell–cell adhesion and are often homotypic, composed only of cancer cells and, in some cases, are admixed with exogenous extracellular matrix and growth factors. However, to best resemble cell–ECM interactions and how they interfere with the cytotoxicity of anti-cancer drugs, induction of endogenous ECM synthesis is preferred.

Recapitulating natural ECM is crucial, as it has been demonstrated that, for example, collagen-rich, TGF-β-dependent ECM is associated with a poor prognosis in many cancer types [3,4,29,30] and the process of ECM secretion, physiologically similar to that occurring in wound healing, poses the basis for angiogenesis, cell proliferation, and immune-cell recruitment, and, in turn, impacts the responsiveness to antineoplastic drugs. Moreover, it must be noted, that mechanical perturbations of the ECM change tumor cell behavior, especially migration and invasion. Hence, ECM deposition is paramount because it resembles the physiological mechanical properties of the tumor [31,32]. Therefore, we decided to use 3D heterotypic models, thus stimulating ECM formation. Indeed, OS cells can reprogram MSCs to a pro-tumorigenic phenotype [3,4] by increasing the metabolic demand of collagen-rich amino acids such as proline and glycine to support ECM production [17,33].

To develop our model, we focused on a panel of OS cell lines, the non-metastatic MG63 and Saos-2 and the metastatic 143B, that are frequently used as a reproducible in vivo model of the disease in immunodeficient mice [18,27,34]. As expected, the presence of MSCs increased, over time, the presence of ECM proteins such as Type I collagen, Type III collagen, and fibronectin [7,8,35,36,37]. The lower observed ECM deposition in 143B spheroids might reflect the propensity of these cells to detach and migrate, eventually seeding to the lungs, whereas non-metastatic cell lines tend to be confined to the primary site, due to the highly intermingled network of ECM proteins. Indeed, the low presence of collagens in 143B led to the formation of spheroids that lacked cohesion, with the cells spreading towards the outer region of the spheroid [38]. Collagens and fibronectins play a fundamental role in OS progression and metastasis by inducing variation in stromal stiffness, inducing the secretion of pro-angiogenetic growth factors, inducing chemoresistance, and facilitating invasion and spread [11,19,22,23,24,25]. Interestingly, fibronectin expression in the TME in patients has also been demonstrated as an unfavorable prognostic factor in a number of malignancies [39,40], including OS [41], and downregulation of specific integrins has demonstrated a possible therapeutic option for the treatment of lung metastasis [42,43].

Indeed, other variations in the expression and arrangement of the ECM have also been reported in cancer patients. TME rearrangements in OS have been correlated with chemotherapy-resistant phenotypes [44], whereas a survival gene signature based on the expression of collagen genes Col3a1, Col4a1, and Col5a2 is correlated with glioblastoma and pancreatic cancer aggressiveness [45,46]. Furthermore, primary breast tumors have been classified based on the ECM composition profile, and this classification correlates with prognosis [47].

By using our model, we could also confirm that, in OS, cancer cells secrete small amounts of ECM proteins, whereas the majority of the secretion derives from the tumor-associated mesenchymal stroma. These results suggest that MSCs may be indirectly responsible for drug resistance through ECM remodeling, collagen crosslinking, and tissue stiffness, which, overall, may contribute to chemoresistance [48]. Moreover, in OS, the spatial proximity of Type I collagen and reactive fibroblast-like cells in the TME was also confirmed in tissue samples of canine OS, whose characteristics are similar to those of the human disease, and in OS xenografts [49,50,51]. The fact that ECM proteins in the context of TME mainly originate from the stromal component of the tumor is also consistent with recent findings: Buschhaus and co-workers, for example, have shown that MSCs reactivate cancer cells to a proliferative state by providing a supporting cancer niche [52,53]. Similarly, in OS, MSCs are recruited by cancer cells and can be found in the bloodstream and at the secondary site, where they contribute to the formation of the metastatic niche in the lung [27,34]. It might be of interest for future studies to analyze the presence of collagen signatures at the metastatic sites [54] and, in other words, to understand whether MSCs home to the lungs prior to cancer cells and provide a hospitable microenvironment by secreting ECM proteins.

We also investigated possible mediators of the cancer–stroma interplay that in OS may contribute to the modulation of ECM deposition and focused on IL-6. In OS TME, IL-6 is extensively produced from cancer-activated MSCs and promotes the expression of target genes related to cell differentiation, survival, apoptosis, proliferation, and stemness [3,27,28]. Briefly, using a monoclonal neutralizing antibody, we demonstrated that IL-6 is responsible for Type I collagen deposition in OS spheroids.

Finally, based on our model, despite its lower relative abundance with respect to other ECM proteins, we found a specific role for Type I collagen in increasing chemoresistance. One possible explanation is that drug distribution in the tumor occurs by diffusion and can be prevented by mechanical obstruction due to stromal stiffness and ECM condensation [7], which, in turn, may also prevent nutrient and oxygen supply, thereby inducing metabolic reprogramming [17] and hypoxic stress [55,56]. As a demonstration, here we showed that by interfering with Type I collagen deposition, OS spheroid sensitivity to DXR is increased. This property is possibly also shared by other components of the ECM in the OS TME, such as Type III collagen and fibronectin [57]. These findings are clinically relevant, as DXR is a first-line agent in OS treatment. Of note, ECM-mediated mechanisms of drug resistance might be extended to other drugs that are currently used in the therapeutic clinical protocols for OS. At this point, it is important to note that chemoresistance is mediated by several molecular mechanisms that are independent of ECM proteins. A reduction in intracellular or intranuclear drug penetration due to lysosomal compartmentalization or acidification of TME [58,59], drug pumping outside the cells via the P-glycoprotein (product of the multidrug resistance gene-1, MDR-1) [60,61], or the transfer of exosomes carrying MDR-1 mRNA and its product P-glycoprotein [62] have all been reported previously and may account for the chemoresistance of metastatic cell lines such as 143B, which express low amounts of the ECM proteins. Altogether, this highlights the importance of developing a model that takes into account different features, such as ECM deposition, and to consider that the observed in vitro effects may be highly context-dependent.

## 5. Conclusions

In conclusion, our findings provide substantial evidence that 3D MSC–OS mixed spheroids can be used to induce the endogenous synthesis of ECM matrix and it can thus be a reliable model for more effective preclinical drug screening. This model also allowed us to demonstrate that MSCs and ECM act as modulators of OS aggressiveness and that IL-6 suppression might be used in combination with standard chemotherapy.

## Figures and Tables

**Figure 1 cancers-15-01221-f001:**
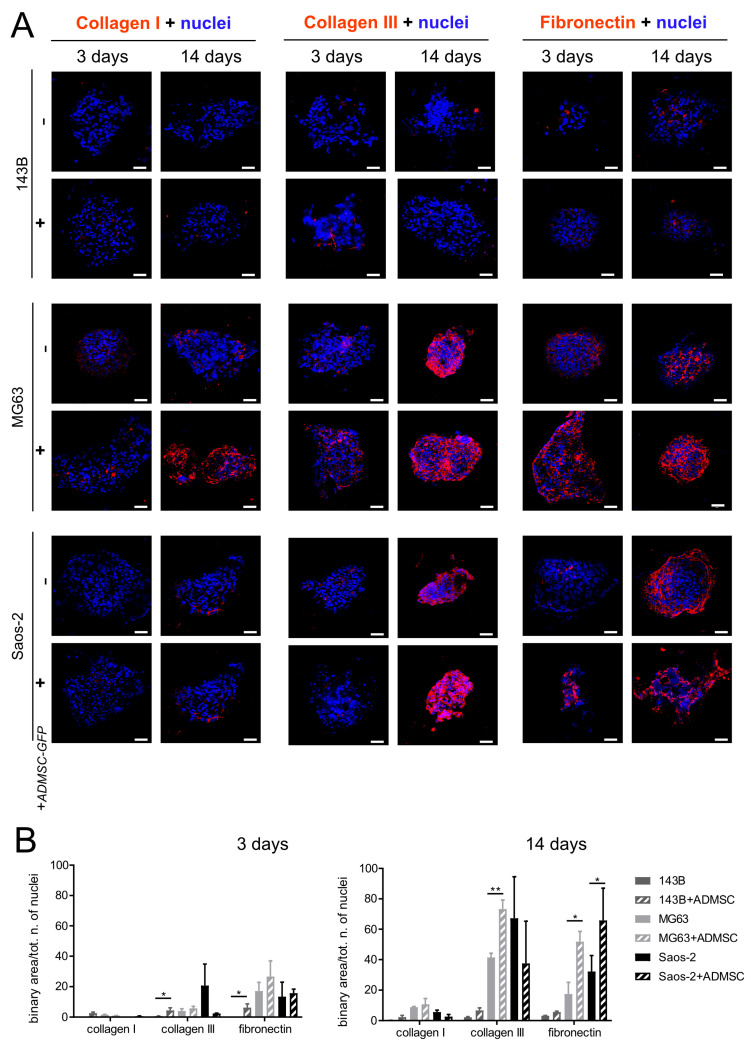
ECM expression of fixed OS spheroids. (**A**) Representative confocal images of OS spheroids obtained with OS cell cultures as homotypic (OS cells alone) or heterotypic (mixed with mesenchymal stromal cells (OS/ADMSC-GFP spheroids)), grown for 3 or 14 days. The spheroids were included in OCT, sliced at 7 μm, and stained for collagen Type I, collagen Type III, and fibronectin with immunofluorescence and counterstained with Hoechst. Images show the merging of the fluorescent signal from the indicated ECM protein (in red) with the fluorescent signal of nuclei (in blue) (scale bar 50 μm); (**B**) Quantification of IF shown in A. The graph expresses the quantification of the total area covered by the fluorescent signal corresponding to the ECM protein (binary area) divided by the total number of nuclei. Data are presented as mean ± SEM of *n* = 10 different fields from 3 independent experiments (unpaired two-tailed Mann–Whitney test; ** *p* < 0.01; * *p* < 0.05).

**Figure 2 cancers-15-01221-f002:**
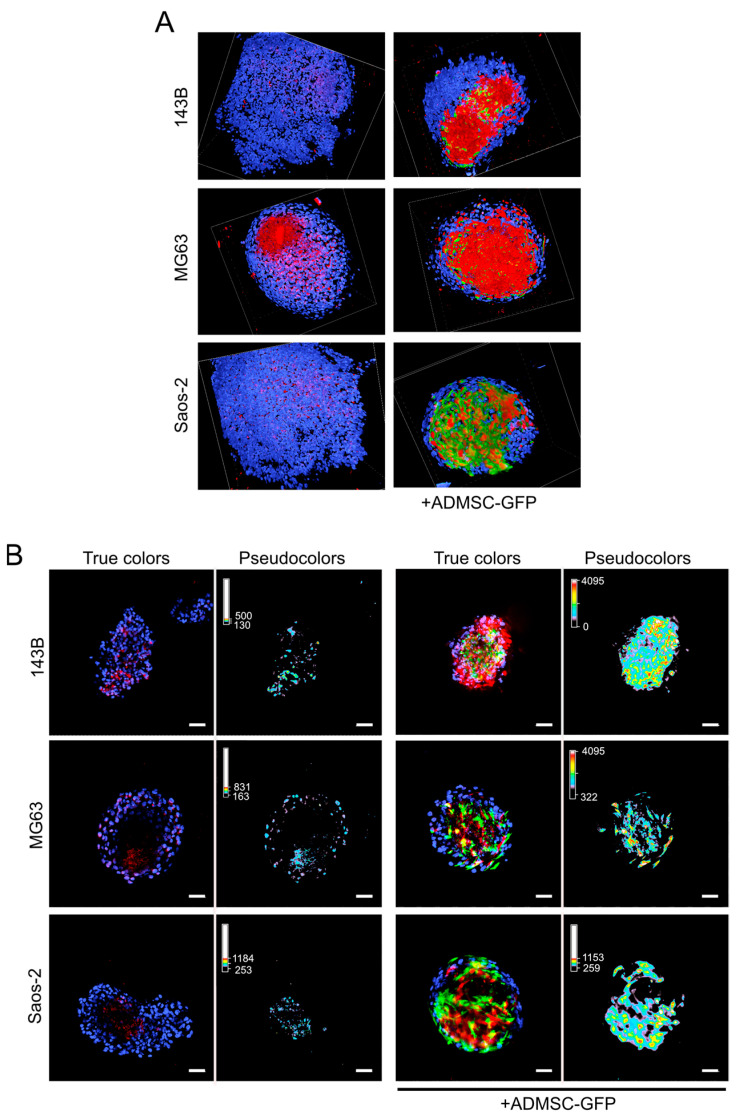
Collagen Type I expression in live OS spheroids. (**A**) 3D volume render after deep Z-scan acquisition with multiphoton confocal microscope (Nikon) of homotypic (only OS cells) and heterotypic (OS/ADMSC-GFP) spheroids from OS cell lines grown for 14 days, live stained for collagen Type I (red), and counterstained for nuclei with Hoechst (blue). Green signal marks the presence of ADMSC-GFP cells. (**B**) Representative Z images of the live spheroids shown in (**A**). Images are a merge of nuclei (blue), collagen Type I (red), and ADMSC-GFP (green). Right panels show collagen Type I intensity profile by pseudo-color staining of the collagen Type I signal. Intensity of the signal is shown on a light blue, green, yellow, and red pseudo-color scale. Scale bar 50 μm.

**Figure 3 cancers-15-01221-f003:**
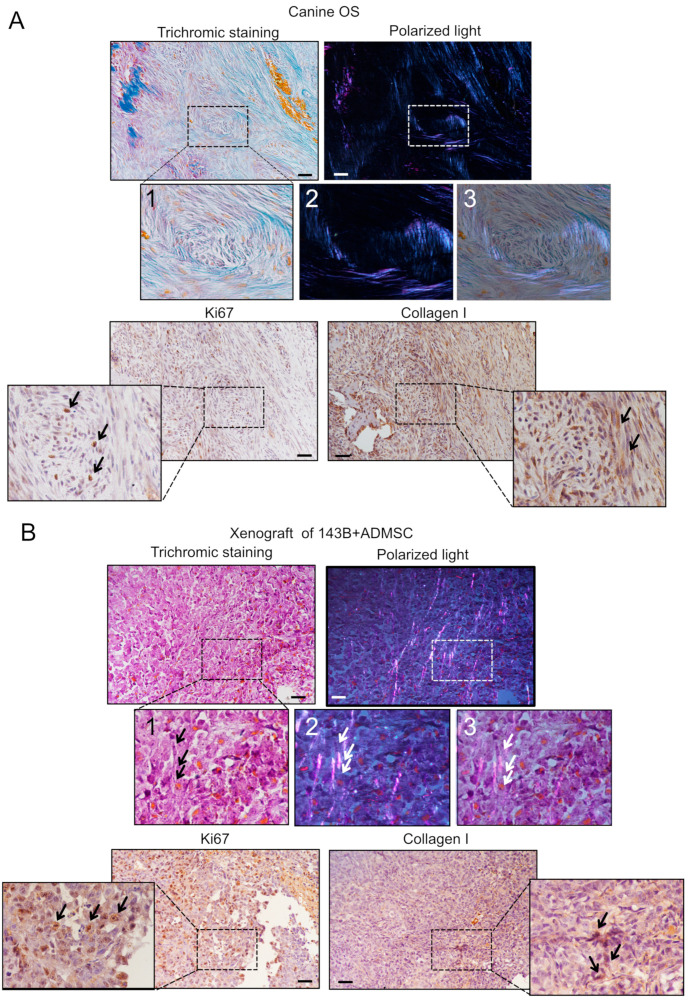
Collagen Type I expression in stromal cells in the OS TME in xenografts and in clinical biopsies from canine patients. Trichromic staining (left) of spontaneous canine OS (**A**) or mouse subcutaneous xenografts of 143B+ADMSC (**B**) were included in paraffin and sliced. The same field shown with trichromic staining was also assessed for fiber orientation using polarized light (right). Type I collagen is positively birefringent with respect to the length of the fibers and appears with blue and purple staining in the trichromic and polarized light images, respectively. Higher magnifications of the dashed squares are shown in panels 1 and 2. In 3, the merge of 1 and 2 is shown. Scale bar 100 μm. Lower panels show immunohistochemistry staining for Ki67 (left panel) and collagen Type I (right panel). Nuclei were counterstained with hematoxylin. Arrows indicate positive staining.

**Figure 4 cancers-15-01221-f004:**
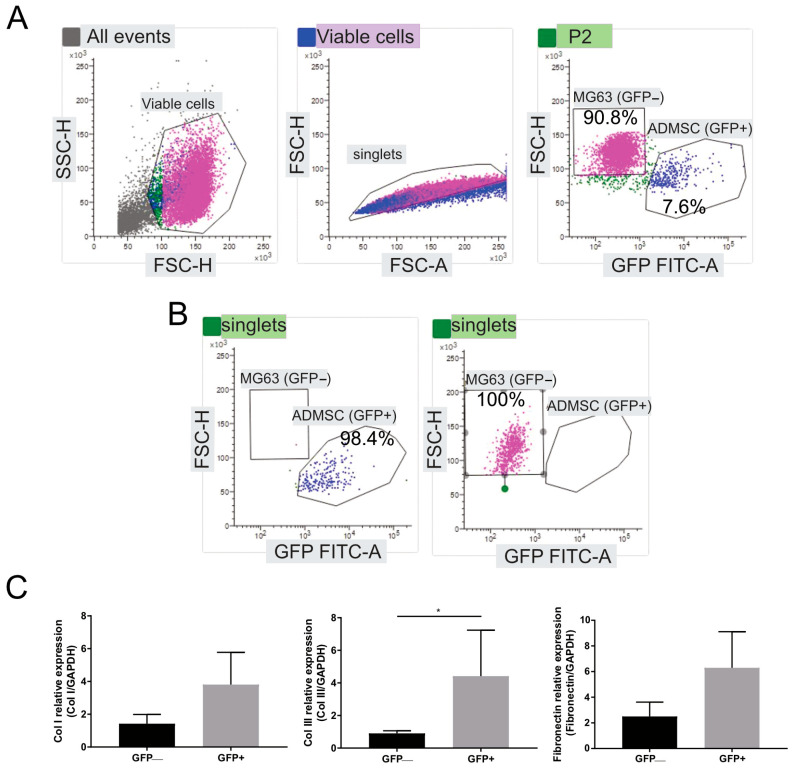
ECM proteins are mainly expressed by ADMSC in the TME. (**A**) MG63/ADMSC-GFP spheroids were grown for 14 days, enzymatically digested, and sorted into GFP-negative OS cell (MG63) and GFP-positive mesenchymal stromal cell (ADMSC) populations. Images show the gating strategy and the percentages of different cells before cell sorting (90.8% for MG63 and 7.6% for ADMSC-GFP). (**B**) Purity assessment of cells after cell sorting. (**C**) Real-time PCR analysis of the indicated genes normalized to GAPDH expression in GFP-negative and GFP-positive cells after cell sorting. Data presented as mean ± SEM of *n* = 3 RNA from 3 independent experiments (unpaired two-tailed Mann–Whitney test; * *p* < 0.05).

**Figure 5 cancers-15-01221-f005:**
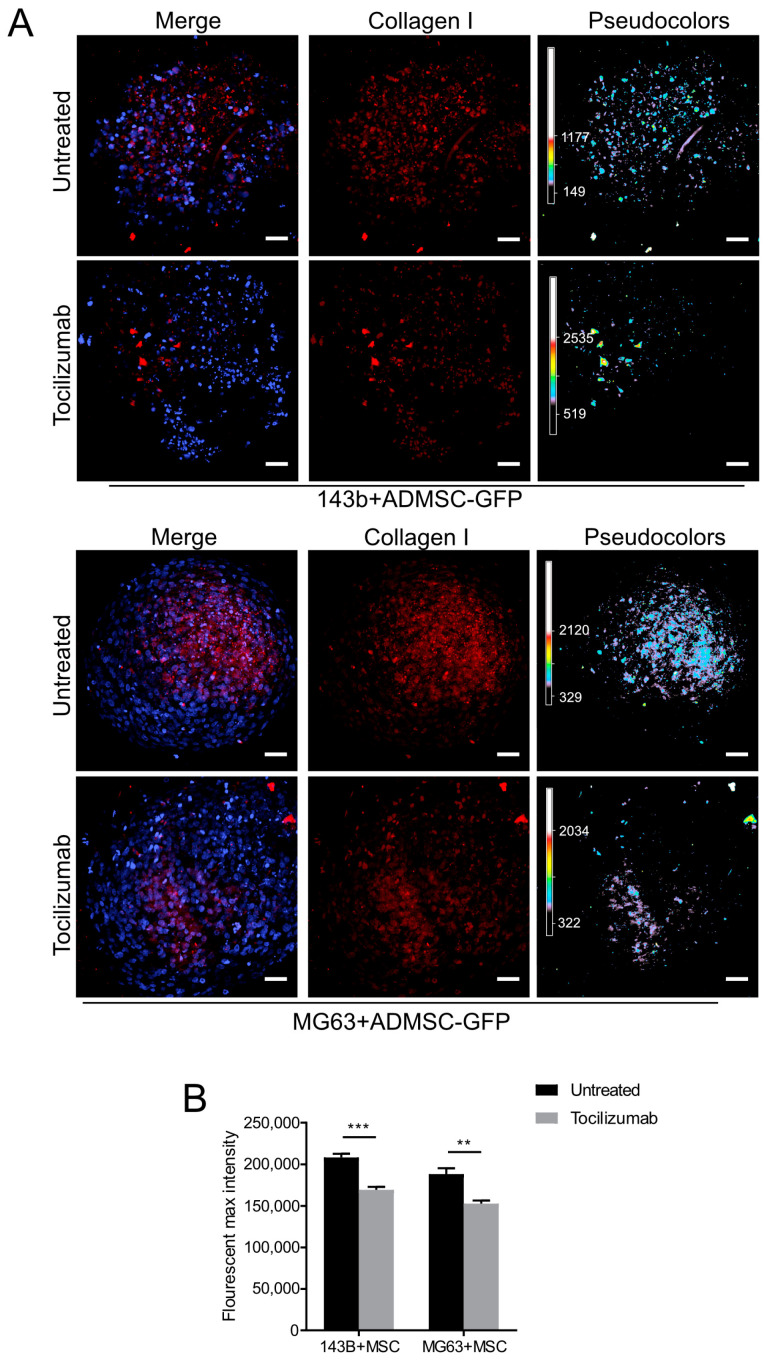
Inhibition of IL-6 secretion decreases collagen Type I expression in OS spheroids. (**A**) Confocal images of spheroids from OS/ADMSC-GFP grown for 14 days, treated with 100 μg/mL of Tocilizumab every 24 h for the whole cell culture period, and then live-stained with immunofluorescence using a specific antibody for collagen Type I (red) and counterstained for nuclei signal with Hoechst (blue). Images show merging of the whole Z-stack (300 μm, stacks every 3 μm); pseudo-colors are used to enlighten the intensity of collagen Type I staining on a light blue, green, yellow, red scale. Scale bar 50 μm; (**B**) Quantification of collagen Type I was performed on the whole Z-stack and expressed as the area covered with the positive red signal for collagen Type I divided by the total number of nuclei. Data presented as mean ± SEM of *n* = 5 of independent experiments (unpaired two-tailed Mann–Whitney test; *** *p* < 0.001; ** *p* < 0.01).

**Figure 6 cancers-15-01221-f006:**
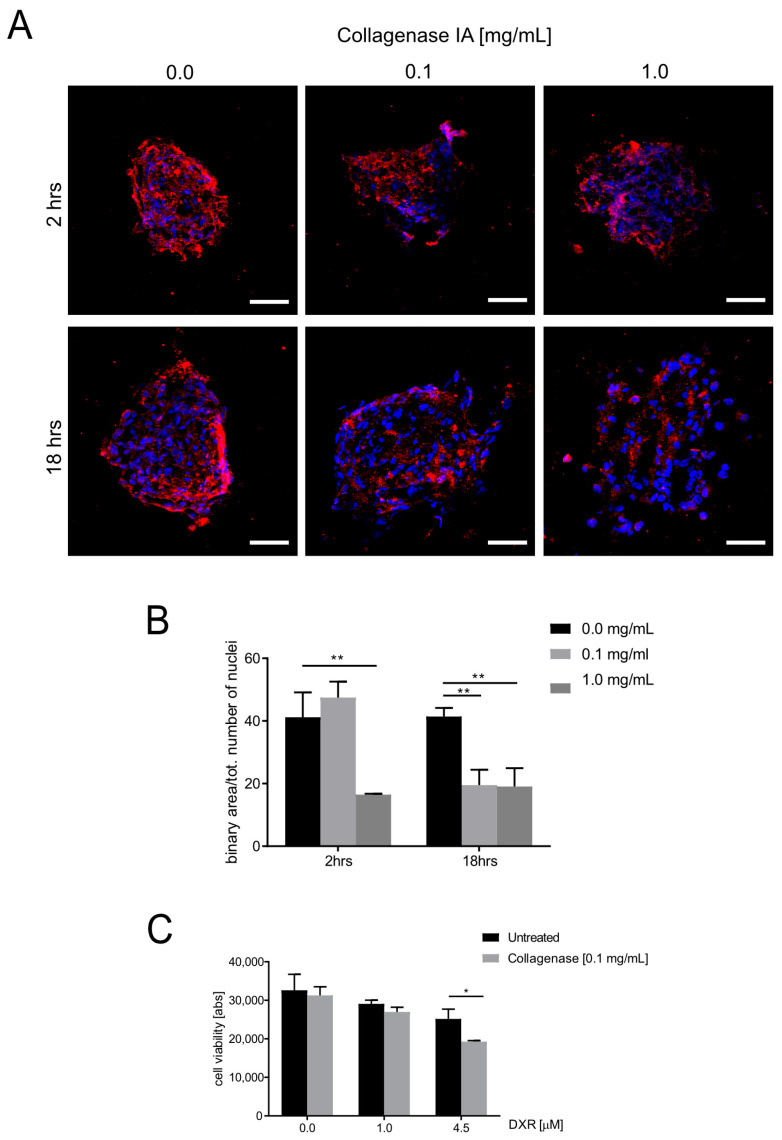
Collagenase treatment sensitizes spheroids to doxorubicin treatment. (**A**) Representative confocal images of spheroids from MG63/ADMSC-GFP grown for 14 days, treated with collagenase IA as indicated, included in OCT, sliced, stained for collagen Type I, and counterstained with Hoechst (blue). Scale bar 50 μm; (**B**) Graph showing quantification of the collagen Type I expression as revealed with an IF assay and expressed as the total area covered by the positive signal for collagen Type I staining divided by the total number of nuclei. Data presented as mean ± SEM of *n* = 10 different fields from 3 independent experiments (unpaired two-tailed Mann–Whitney test; ** *p* < 0.01; (**C**). (**C**) Spheroids from MG63/ADMSC-GFP were grown for 14 days, treated with collagenase IA at 0.1 mg/mL for 18 hrs and next treated with DXR 1 μM or 4.5 μM for 72 h. Viability was assessed with Alamar Blue (unpaired two-tailed Mann–Whitney, * *p* < 0.05).

**Table 1 cancers-15-01221-t001:** IC50 values expressed as mean ± SEM of 2D or 3D cultures of OS cell lines ± ADMSC-GFP treated with doxorubicin.

IC50	143B	143B+ADMSC-GFP	MG63	MG63+ADMSC-GFP	Saos-2	Saos-2+ADMSC-GFP
2D	0.015 ± 0.003	0.026 ± 0.014	0.017 ± 0.021	0.029 ± 0.016	0.014 ± 0.009	0.039 ± 0.236
3D	0.013 ± 0.004	0.020 ± 0.004	0.089 ± 0.004	0.120 ± 0.002	0.072 ± 0.002	0.232 ± 0.054

## Data Availability

Raw data that support the findings of this study are available upon reasonable request from the authors.

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
