# Peer review of "Endogenous Extracellular Matrix Regulates the Response of Osteosarcoma 3D Spheroids to Doxorubicin"

_cancers, 2023, doi:10.3390/cancers15041221_

Round 1
Reviewer 1 Report
The manuscript “Endogenous extracellular matrix regulates the response of os3 teosarcoma 3D spheroids to doxorubicin” by Margherita et al. study aims at generating a 3D model which recapitulates interactions of cancer cells with ECM 20 components and with non-tumor stromal cells and at elucidating the role of ECM deposition in 21 chemotherapy respons.
This paper is well written, clear and the conclusions are supported by the results. However, some corrections are needed in order to improve the overall quality of the manuscript.
1. Please provide a scheme or mechanism as first figure to enhance readability.
2. The introduction is too long, the author's should reduce the introduction.
3. Figure-1 legends are hard to understand especially? What is (c) 20 %?
4. Why the author's select two time soint, 3 and 14 days, I'm wondering that what is happening at Day 7, or day 10? please include the teh dat at Day 10.
5. Figure 2, 3 and 4 need to be reorganized and attract readers with propr labeling.
6. The total quality of the writing poor, please check the typo and grammatical errors.
Author Response
The manuscript “Endogenous extracellular matrix regulates the response of os3 teosarcoma 3D spheroids to doxorubicin” by Margherita et al. study aims at generating a 3D model which recapitulates interactions of cancer cells with ECM 20 components and with non-tumor stromal cells and at elucidating the role of ECM deposition in 21 chemotherapy respons.
This paper is well written, clear and the conclusions are supported by the results. However, some corrections are needed in order to improve the overall quality of the manuscript.
>We thank the reviewer for appreciating our work.
- Please provide a scheme or mechanism as first figure to enhance readability.
>The manuscript presents a graphical abstract that will hopefully help readability
- The introduction is too long, the author's should reduce the introduction.
>We agreed with the reviewer. We have provided a shorter and hopefully clearerversion of it.
- Figure-1 legends are hard to understand especially? What is (c) 20 %?
>We have modified the figure legends which are now hopefully clearer.
- Why the author's select two time soint, 3 and 14 days, I'm wondering that what is happening at Day 7, or day 10? please include the teh dat at Day 10.
>We have added IF and quantifications at 7 days as a new Figure S1; Day 7 shows indeed an intermediate phenotype between day 3 and day 14, suggesting progressive ECM synthesis and deposition consistently over time.
- Figure 2, 3 and 4 need to be reorganized and attract readers with propr labeling.
>Figures have been reorganized as requested.
- The total quality of the writing poor, please check the typo and grammatical errors.
>We have carefully re-read the whole manuscript and corrected all errors. The manuscript is hopefully now more clear.
Reviewer 2 Report
Report on paper "Endogenous extracellular matrix regulates the response of osteosarcoma 3D spheroids to doxorubicin
by Cortini et al.
The paper describes the presence of extracellular matrix (ECM) during formation/growth of spheroids. These spheroids are made of osteosarcoma cells with possible addition of mesenchymal stromal cells (MSC). It is known that in these systems, ECM - in particular collagen - is a very important issue for the development of cancers, as it may be used by cells to migrate/grow on a scaffold. Thus this question is very important, and the role of MSCs can be essential. The authors use confocal microscopy, xenographs/biopsies and RNA isolation/gene expression to analyze the development of endogenous ECM (collagen I, III and fibronectin) using 3 types of cells. The main results are :
- Endogenous collagen deposition is shown for 3 and 14 days of culture
- The role of MSCs is shown to account for ECM deposition
- Collagen deposition is shown to be regulated by IL-6
- ECM influences doxorubicin cytotoxicity in OS spheroids, as shown using collagenase
In general, the paper is quite well written, and results are important for the understanding of tumour stroma, and relevant to future therapies. Results are convincing and well discussed. Before the paper can be accepted, there are a few questions that the authors should address, based on previous studies.
GENERAL QUESTIONS
* Additional references on the role of collagen could be included in the introduction and further discussed :
- Levantal et al. Soft Matter 2007, Mierke et al. Rep. Progr. Physics 2019: Cells can interact with the ECM and align fibers. Could this be observed here ?
- Provenzano et al., BMC Medicine, 2006: Interactions between stromal collagen and tumours. How is such as study relevant to the present one ?
- Tsvirkun et al. J. Biomech. 2022: Collagen affects mechanical properties of spheroids. Can the authors comment on this and infer further ideas on this question, based on their confocal imaging ?
- Whatcott et al. Clin. Cancer Res. 2015: Correlation between patient survival and the presence of collagen. Does this match the conclusions of the present work ?
* The discussion does not account for the role of time in the formation of spheroids and ECM deposition. Could the authors include a few comments about the difference between 3 and 14 days of culture ?
* The authors discuss adhesion, but do not go any further. What are the major adhesion molecules involved here ? What is their role ? What receptors on the Osteosarcoma cells interact with collagen I, III and fibronectin ? Are these integrins ? Maybe a further paragraph about this could be included in the discussion
MINOR POINTS
* Figure legends could be longer to explain better (for a non specialist ) what is shown. In particular Figure 3
* Figure 4 is not large enough and is difficult to read
Author Response
Report on paper "Endogenous extracellular matrix regulates the response of osteosarcoma 3D spheroids to doxorubicin
by Cortini et al.
The paper describes the presence of extracellular matrix (ECM) during formation/growth of spheroids. These spheroids are made of osteosarcoma cells with possible addition of mesenchymal stromal cells (MSC). It is known that in these systems, ECM - in particular collagen - is a very important issue for the development of cancers, as it may be used by cells to migrate/grow on a scaffold. Thus this question is very important, and the role of MSCs can be essential. The authors use confocal microscopy, xenographs/biopsies and RNA isolation/gene expression to analyze the development of endogenous ECM (collagen I, III and fibronectin) using 3 types of cells. The main results are :
- Endogenous collagen deposition is shown for 3 and 14 days of culture
- The role of MSCs is shown to account for ECM deposition
- Collagen deposition is shown to be regulated by IL-6
- ECM influences doxorubicin cytotoxicity in OS spheroids, as shown using collagenase
In general, the paper is quite well written, and results are important for the understanding of tumour stroma, and relevant to future therapies. Results are convincing and well discussed. Before the paper can be accepted, there are a few questions that the authors should address, based on previous studies.
> We thank the reviewer for the general comments and are happy that the work was appreciated. We also thank the reviewer for the interesting and stimulating discussion that his/her comments allowed.
GENERAL QUESTIONS
* Additional references on the role of collagen could be included in the introduction and further discussed :
- Levantal et al. Soft Matter 2007, Mierke et al. Rep. Progr. Physics 2019: Cells can interact with the ECM and align fibers. Could this be observed here?
> It is true that the mechanical properties of the ECM are paramount for the interaction of the cells with the proteins and, hence, for migration and invasion. And the reviewer is right that this phenomenon has been observed in a number of cancers. Unfortunately, we could not observe such mechanism in the experiments we performed. Our lab has been in the past years focused on the study of the acidic tumor microenvironment, which generates aggressive tumor subpopulations that increase migration and invasion. It is also known that tumor acidosis increases the alignment of the fibers and it will be of interest, in the future, for our lab to assess differences in the alignment in different conditions of the tumor microenvironment such as acidosis. The references have been added to the text.
- Provenzano et al., BMC Medicine, 2006: Interactions between stromal collagen and tumours. How is such as study relevant to the present one ?
>This issue is of huge interest. The example provided in the indicated reference is somewhat different from osteosarcoma. Specifically, breast cancer primary tumors occur in a soft tissue, where the generation of collagenic signatures might be easy to be identified. The primary site of osteosarcoma is instead bone; OS is notable for its hallmark production of rich extracellular matrix (ECM) of osteoid (a mix of proteins, including collagen, embedded in a calcified hard matrix) but such a physiological abundance can be somewhat confounding to base a diagnosis on. In other words, a collagenic signature is already physiologically present, meaning that variations occurring during cancer growth might be too low to be visible. Instead, we did hypothesize that MSC might home to the lungs prior to cancer cells and, by collagen deposition, induce the formation of a hospitable metastatic niche. Unfortunately, we tried to perform immunohistochemistry on lungs to assess whether metastasis were surrounded by collagen, but we did not succeed, possibly because 143B cells, in mice, proliferate too quickly and do not allow MSC enough time for ECM deposition in an amount that could be detected by an immunochemical assay. The reference was added to the discussion.
- Tsvirkun et al. J. Biomech. 2022: Collagen affects mechanical properties of spheroids. Can the authors comment on this and infer further ideas on this question, based on their confocal imaging?
> A brief description of the importance of the mechanical properties has been added when commenting Levantal et al. Soft Matter 2007, Mierke et al. Rep. Progr. Physics 2019. Here, it is interesting to note that the authors of the indicated reference observe the same phenomenon we have seen with OS cell lines: 143B, which have low ECM deposition have cells which are more spread towards the outside of the spheroid, whereas MG63 and Saos-2 are more compact and cohesive. The reference was added to the text.
- Whatcott et al. Clin. Cancer Res. 2015: Correlation between patient survival and the presence of collagen. Does this match the conclusions of the present work?
> To our knowledge, no correlation collagen-survival studies on osteosarcoma have been reported. As mentioned before, the high abundance of collagen at the primary site in the bone may result confounding to assess whether OS patients have more or less. The reference has been discussed in the text.
* The discussion does not account for the role of time in the formation of spheroids and ECM deposition. Could the authors include a few comments about the difference between 3 and 14 days of culture ?
> We added images and quantifications of ECM proteins after 7 days of culture as new figure S1. It is now clearer, that ECM deposition occurs slowly but consistently over time, as the quantification at 7 days is intermediate between 3 and 14 days. We have cleared this detail in the text.
* The authors discuss adhesion, but do not go any further. What are the major adhesion molecules involved here ? What is their role ? What receptors on the Osteosarcoma cells interact with collagen I, III and fibronectin ? Are these integrins ? Maybe a further paragraph about this could be included in the discussion
> The questions raised by the reviewer are of high interest and indeed details about the role of integrins have not been discussed in the text. However, we feel that adding a specific paragraph in the discussion might be out-of-topic as the role of integrins has not been assessed during experiments. Nevertheless, we have added some references in the discussion that raise the possibility to use integrin downregulation to reduce metastatic OS spread.
MINOR POINTS
* Figure legends could be longer to explain better (for a non specialist ) what is shown. In particular Figure 3
> The figure legends have been extended as requested.
* Figure 4 is not large enough and is difficult to read
> The labels have be re-written and the figure is hopefully now more visible and clear.